# ListenFormer: Responsive Listening Head Generation with Non-autoregressive Transformers

## ABSTRACT

As one of the crucial elements in human-robot interaction, responsive listening head generation has attracted considerable attention from researchers. It aims to generate a listening head video based on speaker's audio and video as well as a reference listener image. However, existing methods exhibit two limitations: 1) the generation capability of their models is limited, resulting in generated videos that are far from real ones, and 2) they mostly employ autoregressive generative models, unable to mitigate the risk of error accumulation. To tackle these issues, we propose Listenformer that leverages the powerful temporal modeling capability of transformers for generation. It can perform non-autoregressive prediction with the proposed two-stage training method, simultaneously achieving temporal continuity and overall consistency in the outputs. To fully utilize the information from the speaker inputs, we designed an audio-motion attention fusion module, which improves the correlation of audio and motion features for accurate response. Additionally, a novel decoding method called sliding window with a large shift is proposed for Listenformer, demonstrating both excellent computational efficiency and effectiveness. Extensive experiments show that Listenformer outperforms the existing state-of-the-art methods on ViCo and L2L datasets. And a perceptual user study demonstrates the comprehensive performance of our method in generating diversity, identity preserving, speaker-listener synchronization, and attitude matching.

## CCS CONCEPTS

• **Information systems → Multimedia content creation**.

## KEYWORDS

listening head generation, video synthesis, transformer

## 1 INTRODUCTION

Communication is indispensable in the process of social interaction, whether in a school setting or in a professional workplace [2, 37, 45]. In face-to-face communication [24], participants take turns playing the roles of speaker and listener to exchange information. The speaker directly transmits information to the listener through verbal expression, while the listener actively considers the information provided by the speaker, decoding it, and offering real-time feedback

*ACM MM, 2024, Melbourne, Australia*
© 2024 Copyright held by the owner/author(s). Publication rights licensed to ACM.
ACM ISBN 978-x-xxxx-xxxx-x/YY/MM
https://doi.org/10.1145/nnnnnnn.nnnnnnn

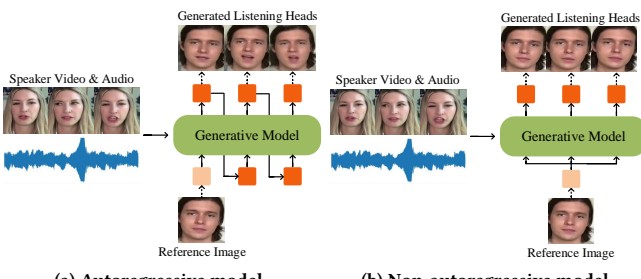

**Figure 1: Concept diagrams of the responsive listening head generation model. Given the speaker inputs and a reference listener image, the autoregressive model relies on past outputs to predict future listener heads, whereas our proposed non-autoregressive model does not depend on previous outputs, deliberately computing results in parallel at each timestep.**

primarily through non-verbal behaviors such as nodding, smiling, headshaking, etc.

The speaker-centric synthesis, specifically talking head generation (THG), has received widespread attention. It plays a significant role in many human-robot interaction (HRI) applications, such as film production, games, and education. Researchers use still images and audio clips to generate vivid speaking videos, advancing towards improving lip-synchronization quality [8, 9, 17], adding emotions [15, 23, 49], and achieving free pose control [28, 31, 56]. However, as another crucial component of HRI, research on responsive listening head generation (LHG), is still in its early stages. The synthesis of smooth listening head videos is also crucial for successful communication [34, 40]. Through real-time feedback, the listener demonstrates their level of engagement in communication, making the conversation easier to understand for both parties. In addition to modeling everyday scenarios, it holds great potential for enriching virtual character modeling, synthesizing fake audiences, and various other applications involving responsive listeners.

Similar to THG, LHG also involves the synthesis of human heads and faces. Therefore, there are many aspects that can be borrowed and applied. For instance, 3D Morphable Models (3DMM) [3, 12, 27] are often used in facial parameters modeling in the THG tasks. Similarly, this approach can be applied to LHG [35, 57] in order to maintain the stability of reconstructed faces. Meanwhile, there are differences between the two. Firstly, THG focuses solely on the speaker, while LHG spans both the speaker and the listener, requiring more consideration of how listening behaviors are influenced by the speaker signals. Additionally, LHG receives signals from both the speaker's audio and video modalities, requiring consideration of the audio-visual fusion issue.

At the earliest, static images, repeated frames, or pre-scripted animations were commonly used to synthesize listeners. However, they often appeared too rigid and were unable to respond realistically to the speaker [57]. Recently, the LHG task was redefined and introduced by Zhou et al. [57], who also curated the audio-visual ViCo dataset comprising video pairs of speakers and listeners. Almost simultaneously, Ng et al. [35] released a novel in-the-wild dataset of dyadic conversations and proposed L2L for understanding human interactional communication. Subsequently, substantial efforts [7, 21, 41, 54] have been devoted to investigating listening head generation techniques. However, these methods still encounter three primary limitations. Firstly, the quality and naturalness of the generated listener videos are currently not good enough. There is still a significant gap compared to real videos, largely due to limitations in model performance. Therefore, it is crucial to have a suitable and effective generative model for this task. Secondly, autoregressive models have inherent limitations. As seen in Fig. 1(a), most existing methods [7, 21, 35, 41, 54, 57] employ autoregressive models, making it difficult to avoid issues such as slow synthesis speed and error accumulation. Moreover, there is insufficient attention to the fusion of audio and video signals for speakers. In LHG, the input speaker signals consist of two modalities: audio and visual motion. Most existing methods [7, 21, 54, 57] simply concatenate the signals from these two modalities, overlooking the importance of cross-modal fusion. A robust fusion method is necessary to better extract representative features from the speaker inputs.

To address the aforementioned problems and meet the multifaceted requirements, we propose a non-autoregressive transformer-based model ListenFormer, which captures the speaker's audio and video signals as well as a reference image to generate highly realistic listener videos. It is important to note that, a two-stage training method is applied to ListenFormer. During the pre-training stage, we employ the teacher-forcing method. This means that real and continuous listener head coefficients were input into the decoder. In the fine-tuning stage, we modify the decoder input to consist of repeated reference image coefficients. As shown in Fig. 1(b), in this way, we achieve the non-autoregressive prediction of ListenFormer. To capture the representative features, we propose a novel audio-motion attention fusion module (AMAF) to embed speaker's audio and motion features. The proposed module utilizes cross-modal attention to discover key information aligned along the temporal sequence. In addition, we experiment with several decoding methods to address the issue of temporal infinite extrapolation for ListenFormer. We conduct extensive experiments on ViCo and L2L datasets and achieve state-of-the-art performance on both datasets. Our code and benchmark will be released.

Overall, our contributions are summarized as follows:

- We propose a transformer-based model ListenFormer that can predict diverse and high-quality listening head videos in a non-autoregressive manner conditioned on the listener's reference image and speaker's audio and motion features.
- The audio-motion attention fusion module (AMAF) is designed to integrate cross-modal features in order to provide representative speaker-related information to the decoder.
- We present an efficient sliding-window decoding method, which addresses the transformer's inability to extrapolate infinitely.

- Experimental results show a significant improvement achieved by our proposed method compared to other state-of-the-art methods on the ViCo and L2L dataset in terms of visual naturalness, generation diversity, identity-preserving, speaker-listener synchronization, and attitude matching.

## 2 RELATED WORK

### 2.1 Responsive Listening Head Generation

In early works, several rule-based methods [5, 6] were employed to produce listener heads. However, those videos fall far short in terms of naturalness and realism. Subsequently, some data-driven approaches [14, 36] based on facial keypoints were used to generate 2D listener motions, but they lost many details of facial expressions.

In recent years, many 3D-based methods have been developed due to their excellent facial reconstruction capabilities. Zhou et al. [57] established a high-quality speaker-listener dataset, named ViCo. The proposed baseline utilizes long-short term memory (LSTM) as the sequential model to handle the input of speaker audio and visual signals, generating facial 3DMM coefficients for the listener. At almost the same time, Ng et al. [35] proposed a novel motion-encoding VQ-VAE [47] to learn a discrete latent representation of realistic listener motion. Later, Huang et al. [21] adopted an enhanced renderer and video restoration module, improving the quality of the generated listening videos. Recently, some methods [7, 55] have attempted to incorporate semantic information into the inputs of the task with the pre-trained language model [26]. However, the methods mentioned above mostly employ autoregressive models, which cannot avoid the issue of error accumulation during the generation process. In contrast, the non-autoregressive prediction approach of Listenformer can largely overcome this limitation.

### 2.2 Transformers in Audio-Visual Learning

Transformer [48] was initially proposed for sequence-to-sequence (seq2seq) translation in the field of natural language processing (NLP). Unlike recurrent neural networks (RNNs) that recursively process sequence tags, transformers can parallelly attend to all tokens in the input sequence, effectively modeling contextual information. The transformers have proven to be a powerful alternative to RNNs in various sequential tasks and have achieved marvelous success in audio-visual learning tasks such as speech recognition [32, 42], emotion recognition [16, 20, 46, 58] and event detection [18, 29, 33]. Some of the most recent works on speech-driven THG [13, 22, 53] have explored the power of transformers in modeling facial features and produced impressive results.

Despite its many advantages, the transformer as a generative model also has notable issues. For instance, the traditional Transformer, being an autoregressive model, suffers from the problem of error accumulation during inference. Additionally, the challenge of temporal infinite extrapolation has been a persistent concern for many researchers working with transformers [1, 43, 44, 52]. After comprehensive consideration, our work relies on a novel non-autoregressive transformer for the 3D reconstruction of the listener's face due to its excellent temporal modeling capability. Moreover, we explore various decoding approaches to address the challenge of infinite extrapolation.

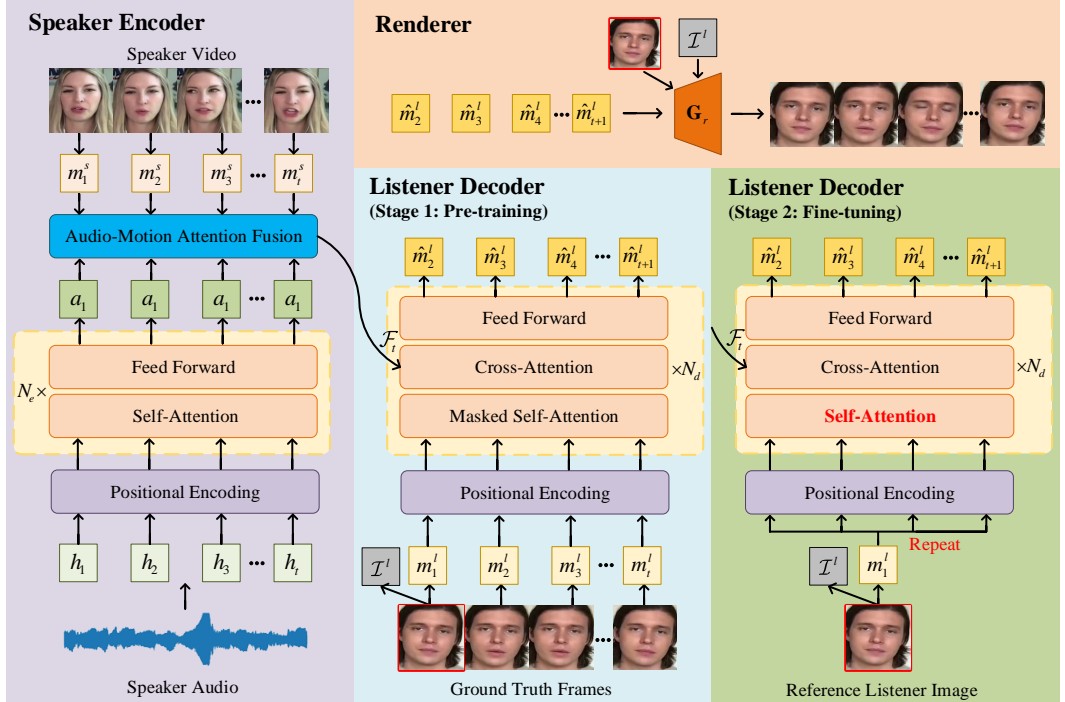

Figure 2: Overview of our proposed ListenFormer. An encoder-decoder model $G_m$ with Transformer architecture takes the speaker's audio $\mathcal{H}_t$ and motion features $\mathcal{M}_t^s$ as well as the reference listener coefficient $m_1^l$ as inputs and generates a sequence of listener coefficient $\hat{\mathcal{M}}_{t+1}^l$, which are fed into a renderer $G_r$ along with the reference listener image $v_1^l$ and identity-dependent coefficients $\mathcal{I}^l$ to produce responsive listening videos $\mathcal{V}_{t+1}^l$. The audio-motion attention fusion module (blue block) is designed for cross-modal robust representation extraction. In addition, the two-stage training method implements non-autoregressive prediction for ListenFormer. $N_e$ and $N_d$ respectively represent the number of layers in the transformer encoder and decoder.

## 3 METHOD

### 3.1 Problem Formulation

We formulate the LHG task as a seq2seq learning problem. Given an input video $\mathcal{V}_t^s = \{v_1^s, \cdots, v_t^s\}$ of a speaker head in timestamps ranging from $\{1, ..., t\}$, containing a corresponding audio signal $\mathcal{S}_t$. The goal here is to produce a model $G$ (Fig. 2) that can synthesize the whole listener's head video sequence $\hat{\mathcal{V}}_{t+1}^l = \{\hat{v}_2^l, \cdots, \hat{v}_{t+1}^l\}$. Formally,

$$\hat{\mathcal{V}}_{t+1}^l = G(\mathcal{V}_t^s, \mathcal{S}_t, v_1^l) \quad (1)$$

where $v_1^l$ denotes the reference head image of the listener.

Following [57], we apply the 3D-based method and divide the $G$ into $G_m$ and $G_r$. As shown in Fig. 2, $G_m$ consists of a speaker encoder and a listener decoder, which is used to predict the 3D reconstruction coefficients of listeners. And $G_r$ is used for 3D face rendering, as depicted in the 'Renderer' part in Fig 2. In the proposed $G_m$, the transformer encoder transforms audio feature $\mathcal{H}_t = \{h_1, \ldots, h_t\}$ into deep representation $\mathcal{A}_t = \{a_1, \ldots, a_t\}$. Meanwhile, we extract the 3D reconstruction coefficients $\{\alpha, \beta, \delta, p, \gamma\}$ which denote the identity, expression, texture, pose, and lighting, respectively. They are split into two components: $\mathcal{I} = (\alpha, \delta, \gamma)$ to represent relatively fixed, identity-dependent coefficients, and $m = (\beta, p)$ to represent relatively dynamic, identity-independent coefficients. These

identity-independent coefficients extracted from speaker videos can be denoted as $\mathcal{M}_t^s = \{m_1^s, \cdots, m_t^s\}$. Then, the audio-motion fusion module fuses $\mathcal{M}_t^s$ and $\mathcal{A}_t$ to get the fusion representation $\mathcal{F}_t$. The transformer decoder receives $\mathcal{F}_t$ and the identity-independent coefficient $m_1^l$ of the reference listener image to non-autoregressively predict the listener coefficients $\hat{\mathcal{M}}_{t+1}^l = \{\hat{m}_2^l, \cdots, \hat{m}_{t+1}^l\}$. We formulate the procedure as:

$$\hat{\mathcal{M}}_{t+1}^l = G_m(\mathcal{M}_t^s, \mathcal{H}_t, m_1^l) \quad (2)$$

Finally, we use the pre-trained rendering model [39] to generate the realistic listening video. Formally,

$$\hat{\mathcal{V}}_{t+1}^l = G_r(\hat{\mathcal{M}}_{t+1}^l, \mathcal{I}^l, v_1^l) \quad (3)$$

where $\mathcal{I}^l$ is the identity-dependent coefficient of the given listener.

For the remainder of this section, we describe each component of the ListenFormer architecture in detail.

### 3.2 Transformer Encoder

We adopt the vanilla Transformer encoder [48]. It is composed of a sinusoidal positional encoding and a stack of sub-layers, converting the audio feature vectors $\mathcal{H}_t$ into contextualized representations $\mathcal{A}_t$. Each encoder layer consists of multi-head self-attention and fully connected feed-forward networks. Note that layer normalization

and residual connection are omitted for simplicity in Fig. 2. The audio representations outputted by the encoder are sent to the audio-motion fusion module.

## 3.3 Transformer Decoder and Two-stage Training

The decoder is also composed of a sinusoidal positional encoding and a stack of sub-layers. Different from the encoder, each decoder layer consists of self-attention, cross-attention, and feed-forward networks. The output identity-independent 3D facial coefficients of listeners are sent to the renderer for video reconstruction.

In the pre-training stage, we apply the teacher-forcing scheme, which is shown in the middle of Fig. 2 (light blue background). At each time step, the decoder receives the real target coefficients $\mathcal{M}_t^l = \{m_1^l, \cdots, m_t^l\}$ (ground truth) along with the fusion representation, instead of using predictions generated by the model itself. This speeds up the training process and minimizes cumulative errors during the training phase. We also apply masks in the self-attention in the first training stage to prevent current output from being affected by subsequent positions according to [48].

Although the teacher-forcing scheme helps the model learn temporal continuity of the output, the model must rely on its own generated previous coefficients and generate predictions autoregressively during the inference phase, leading to inconsistency between training and inference. Therefore, we modify the input of the decoder in the fine-tuning stage. Specifically, as shown in the right of Fig. 2 (light green background), the 3D coefficient $m_1^l$ of the reference listener image is replicated along the time axis and inputted into each time step to replace the ground truth $\mathcal{M}_t^l$. As shown in Fig. 1(b), the model can perform non-autoregressive inference consistent with the training phase, thereby avoiding the issue of cumulative errors. Additionally, such approach does not require masks in self-attention and tends to provide a globally consistent motion.

On the one hand, the first teacher-forcing pre-training stage forces the model to optimize in the right direction in the early stages of training. During the experimental phase, we find that skipping the pre-training stage and directly proceeding to the non-autoregressive training in the second stage does not yield more satisfactory results. For more details, refer to Section 4.5.2. This indicates the importance of the pre-training stage for the final performance of ListenFormer. On the other hand, the prediction approach in the second fine-tuning stage is the non-autoregressive method we ultimately aim for in inference. Since people usually do not make large head movements during the listening process, using repeated reference frames as the input for the decoder helps maintain the stability of the predictions. As a result, the combination of the two training stages allow ListenFormer to simultaneously learn temporal continuity and overall consistency, achieving a good balance between facial motion diversity and stability.

Once the complete 3D facial coefficient sequence is produced, the model is trained by minimizing the regression loss between the

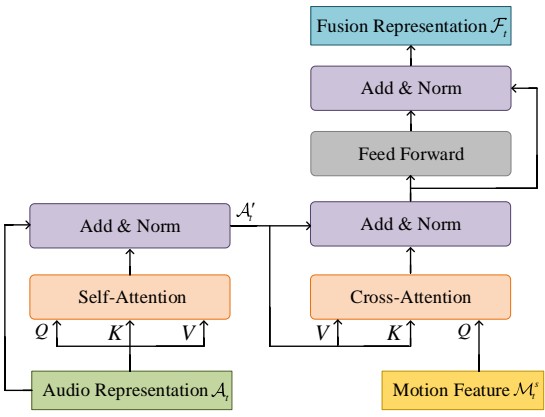

**Figure 3: Structure of the AMAF module.**

decoder outputs and ground truths, which is calculated as:

$$
\begin{aligned}
L = \sum_{t=2}^{T} ||\beta_t^l - \hat{\beta}_t^l||_2 + ||p_t^l - \hat{p}_t^l||_2 \\
+ \sum_{t=2}^{T} w_1 ||\mu(\beta_t^l) - \mu(\hat{\beta}_t^l)||_2 + \sum_{t=2}^{T} w_2 ||\mu(p_t^l) - \mu(\hat{p}_t^l)||_2
\end{aligned}
\tag{4}
$$

where $\hat{\beta}_t^l$ and $\hat{p}_t^l$ represent the generated expression and pose coefficients of listeners, respectively. The last two terms of Eq. 4 are applied to guarantee the inter-frame continuity, where $\mu(\cdot)$ measures the inter-frame changes. $w_1$ and $w_2$ are the adjustable parameters for different losses.

## 3.4 Audio-Motion Attention Fusion

When we listen to others, the speaker's audio, facial expressions, and head motions can convey messages to us. Often, there is a strong correlation between them. Therefore, it is crucial to effectively utilize multi-modal features (audio and motion) in the LHG task. The previous works [7, 21, 54, 57] only concatenated audio and motion features, which is a coarse fusion fashion. Here, we design a novel AMAF module for finer interaction, as shown in Fig. 3.

In our view, audio conveys richer information in communication compared to motion, for example, semantic information is lacking in motion. In cross-modal fusion, we prioritize audio as the primary information stream, with motion serving as supplementary modality. Experimental results in Section 4.5.1 demonstrate that this approach performs better than regarding motion as the primary modality. Before the cross-modal fusion, to model the temporal relations of audio representation $\mathcal{A}_t$, we feed it into a multi-head self-attention module. Then, enhanced representation $\mathcal{A}_t'$ interacts with motion representation $\mathcal{M}_t^s$ in a cross-attention way, where the queries are from $\mathcal{M}_t^s$, keys and values are from $\mathcal{A}_t'$, respectively:

$$
\text{Interact}(\mathcal{A}_t', \mathcal{M}_t^s) = \text{Softmax}\left(\frac{\mathcal{M}_t^s W^Q \cdot (\mathcal{A}_t' W^K)^{\mathsf{T}}}{\sqrt{d}}\right) \mathcal{A}_t' W^V
\tag{5}
$$

where $W^Q$, $W^K$, $W^V$ are learnable parameters and $d$ is a scaling factor.

We expect that the motion feature plays a role as queries in the cross-attention mechanism, strengthening the audio representation closely associated with head motion and facial expressions to obtain a more comprehensive fused representation $\mathcal{F}_t$. At last, a feed-forward layer is applied to output the fusion representation. Residual connection and layer normalization are employed after attention and feed-forward layers to ensure the training stability of the fusion process.

## 3.5 Decoding

Because there is a significant disparity in the lengths of videos in the training datasets (ranging from 1 to 71 seconds), the video clips are divided into fixed-length segments for training. This results in limited generalization ability of the transformer during the decoding phase for longer sequences. Methods to enhance the length extrapolation ability of transformers have garnered widespread attention. Existing approaches mainly fall into relative position encoding [38, 43, 44], context window extension [1, 10, 50], and so on. There is, however, a scarcity of methods specifically addressing seq2seq tasks. To address this issue for non-autoregressive ListenFormer, we explore three different decoding methods, corresponding to the three subfigures in Fig. 4.

Fig. 4(a) represents the **all-in** decoding method, where the entire clip with length $T$ is inputted at once, and all predictions are generated in a single decoding step. This method not only results in high computational complexity $O(T^2)$ but also yields poor performance due to limitations in extrapolation ability. Fig. 4(b) represents the **step-by-step** decoding method. The input segment has a fixed window length $L$ and only one frame is slid in at each step. Meanwhile, the output of the last frame of each segment is concatenated to the final predictions. Although its computational complexity is reduced compared to the all-in approach, its $O(TL^2)$ complexity can considerably slow down the decoding process when dealing with long videos. Furthermore, while this stepping approach performs well in autoregressive large language models (LLMs) [52], it is not suitable for ListenFormer which computes all inputs' results in parallel. While each input is step-by-step, for non-autoregressive inference, even small changes in input can result in non-coherent outputs between each step. Therefore, the step-by-step method may lead to significant temporal jitter in the final output.

Fig. 4(c) represents the proposed **sliding window with a large shift** decoding method. Similar to the step-by-step method, the input window length $L$ remains fixed, but the sliding shift $S$ is expanded to approach the size of the window length $L$. A slight overlap helps to smooth the output at the junctions of segments. The outputs of non-overlapping frames are concatenated to the final prediction at each step. This method not only further reduces the complexity to $O(TL)$, but also alleviates the jitter issue associated with the stepping method. To further maintain the predictions coherence, in the (b) and (c) methods, the reference image is replaced by the output of the beginning frame taken from the previous segment after the first step. More performance comparisons can be found in Section 4.5.3.

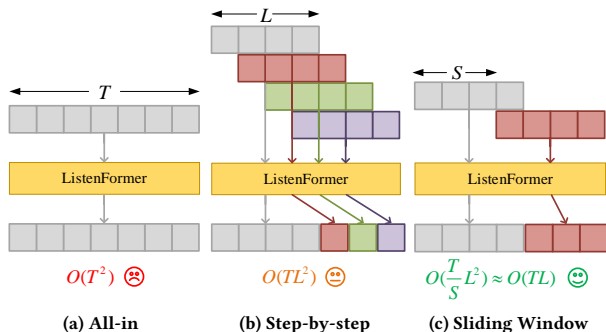

(a) All-in     (b) Step-by-step     (c) Sliding Window

**Figure 4: Illustration of three decoding methods. The non-autoregressive ListenFormer, trained on inputs of length $L$, predicts the outputs of length $T$ ($T \gg L$). And the shift length in (c) is $S$ ($S \approx L$).**

## 4 EXPERIMENT

### 4.1 Experimental Settings

*4.1.1 Dataset.* We train and validate our model on two conversation portrait datasets, the ViCo [57] and L2L [35] datasets. The ViCo dataset contains 483 video clips ranging from 1 to 71 seconds. Specifically, it includes the identities of 76 listeners and 67 speakers, and each clip contains face-to-face interaction between two realistic subjects. It is divided into training $\mathcal{D}_{train}$, test $\mathcal{D}_{test}$, and out-of-domain $\mathcal{D}_{ood}$ subsets. All identities present in $\mathcal{D}_{test}$ are also found in $\mathcal{D}_{train}$, while identities in $\mathcal{D}_{ood}$ do not overlap with those in $\mathcal{D}_{train}$. The L2L dataset is a 72-hour versus 95-minute dataset collected in the wild, which comes from YouTube with six identities. Each video features a plethora of interviewees and hosts from a variety of backgrounds. Note that the L2L dataset only provides 3D expression and pose coefficients along with corresponding speaker-only audio features, and does not include the original videos.

*4.1.2 Evaluation Metrics.* On the ViCo dataset, both the feature-level and video-level metrics are applied for comprehensive comparison. For the former one, the L1 distance is employed to represent the disparity between the predicted angles, expressions, translation coefficients, and the ground truth. Angle and translation coefficients are two components that constitute the pose coefficient $p$. For the latter one, we adopt Fréchet Inception Distance (FID) [19], Structural Similarity (SSIM) [51], Peak Signal-to-Noise Ratio (PSNR), and Cumulative Probability of Blur Detection (CPBD) [4]. Additionally, to evaluate identity preservation, we utilize cosine similarity (CSIM) between identity features extracted from ArcFace [11] on generated and source videos.

On the L2L dataset, due to the unavailability of the original videos, only feature-level metrics (L1 distance and Fréchet distance (FD)) for the expression and angle coefficients are applied.

*4.1.3 Comparison Methods.* Five state-of-the-art responsive listener head generation methods are selected as comparing methods. **ViCo** [57] utilizes an LSTM-based sequential decoder to predict the pose and expression features of the listener subject. **PCHG** [21]

**Table 1: The L1 Distance (×100) of different listening head generation methods on ViCo dataset. Each cell in the table represents the feature distance of angle/expression/translation coefficients respectively. Lower is better. The bold and underlined notations represent the Top-2 results. The ∗ indicates that we directly follow the official report results of MFR-Net, while the results of other comparison methods are reproduced on our own system.**

| Method | Testset | Positive | | | Neutral | | | Negative | | | Average | | |
|---|---|---|---|---|---|---|---|---|---|---|---|---|---|
| | | Angle | Exp | Trans | Angle | Exp | Trans | Angle | Exp | Trans | Angle | Exp | Trans |
| ViCo [57] | $\mathcal{D}_{test}$ | 7.22 | 14.66 | 6.16 | 5.33 | 12.87 | 6.80 | 13.86 | 17.73 | 6.96 | 9.53 | 15.57 | 6.59 |
| | $\mathcal{D}_{ood}$ | 8.45 | 16.68 | 7.05 | 7.05 | 15.17 | 6.38 | 6.85 | 17.66 | 6.96 | 7.54 | 16.41 | 6.79 |
| PCHG [21] | $\mathcal{D}_{test}$ | 7.24 | 14.71 | 6.06 | 5.32 | 12.90 | 7.37 | 13.84 | 17.94 | 6.89 | 9.53 | 15.68 | 6.60 |
| | $\mathcal{D}_{ood}$ | 8.42 | 16.73 | 7.05 | 7.01 | 15.32 | 6.79 | 6.86 | 17.70 | 6.81 | 7.51 | 16.50 | 6.90 |
| DSPN [54] | $\mathcal{D}_{test}$ | 4.82 | 5.80 | 12.89 | 5.32 | 11.84 | 5.71 | 14.39 | 17.83 | 7.74 | 8.71 | 14.67 | 6.55 |
| | $\mathcal{D}_{ood}$ | 7.69 | 15.77 | 7.08 | 6.30 | 13.11 | 6.20 | 7.76 | 14.58 | 6.37 | 7.23 | 14.53 | 6.58 |
| MFR-Net* [30] | $\mathcal{D}_{test}$ | 5.36 | 13.73 | 5.94 | 5.35 | 12.32 | **4.58** | **11.78** | **13.46** | **5.48** | **6.82** | 13.37 | 6.02 |
| | $\mathcal{D}_{ood}$ | 9.03 | 13.72 | 6.29 | 6.27 | 12.96 | **4.77** | 7.77 | 15.51 | **5.78** | 8.12 | 14.70 | 6.37 |
| Ours | $\mathcal{D}_{test}$ | **4.24** | **11.61** | 5.62 | **3.30** | **9.25** | 4.89 | 12.47 | 17.04 | 6.49 | 7.35 | **13.36** | 5.84 |
| | $\mathcal{D}_{ood}$ | **4.89** | **13.63** | 5.94 | **3.72** | **12.09** | 5.62 | **6.23** | **12.92** | 6.51 | **4.95** | **12.90** | 5.98 |

**Table 2: Quantitative results on video-level metrics with different methods on ViCo dataset. The upward arrow indicates that higher values correspond to better results, while the downward arrow indicates the opposite.**

| Method | SSIM ↑ | CPBD ↑ | PSNR ↑ | FID ↓ | CSIM ↑ |
|---|---|---|---|---|---|
| ViCo [57] | 0.57 | 0.16 | 17.34 | 27.03 | 0.49 |
| PCHG [21] | 0.56 | 0.16 | 16.79 | 26.57 | 0.49 |
| DSPN [54] | 0.59 | 0.15 | 17.64 | 26.33 | 0.58 |
| MFR-Net* [30] | 0.59 | **0.18** | 17.82 | **20.08** | - |
| Ours | **0.62** | 0.17 | **18.89** | 24.52 | **0.63** |

**Table 3: Quantitative results on feature-level metrics with different methods on L2L dataset.**

| Method | Expression | | Angle | |
|---|---|---|---|---|
| | L1 ↓ | FD ↓ | L1 ↓ | FD ↓ |
| ViCo [21] | 30.28 | 15.08 | 7.15 | 6.77 |
| DSPN [54] | 23.65 | 3.16 | 5.82 | 1.54 |
| L2L [35] | 37.22 | 17.6 | 9.90 | 8.13 |
| Ours | **10.45** | **2.66** | **2.71** | **1.33** |

modifies the post-processing approach during the rendering process based on ViCo. **DSPN** [54] is a dual-stream prediction network, which consists of LSTMs and temporal convolutional networks (TCN) [25]. **MFR-Net** [30] employs the probabilistic denoising diffusion model to predict multi-faceted response. **L2L** [35] learns a realistic manifold of listener motion through a novel sequence-encoding.

*4.1.4 Implementation Details.* On the ViCo dataset, the input video frames are cropped to 256 × 256 size at 30 FPS and the audio signals are extracted into 45-dimensional acoustic features, including mel-frequency cepstral coefficients (MFCC), energy, zero-crossing rate (ZCR), and loudness. The window length of the speaker clip is set to be 90 frames with a shift of 80 frames. The 3DMM coefficients are extracted with the guides of PIRender [39]. The identity-dependent features are in $\mathbb{R}^{187}$, and the identity-independent features are in $\mathbb{R}^{70}$.

On the L2L dataset, the audio signals are extracted into 128-dim mel features. Following [35], the parameters representing identity-independent features include 50 expression coefficients along with a 3D jaw rotation, as well as 3D head rotation in Euler angles.

As for model details, we utilize 3 transformer encoder layers and 3 transformer decoder layers along with 4 attention heads. Due to the lack of original videos, the rendering part is not required when conducting experiments on the L2L dataset.

## 4.2 Quantitative Evaluation

*4.2.1 ViCo dataset.* Tab. 1 shows the feature-level metrics on the $\mathcal{D}_{test}$ and $\mathcal{D}_{ood}$ subsets of the ViCo dataset, through evaluations conducted on generated angle, expression, and translation features. Following [57] and [30], results are presented for three different attitudes, along with their average values. Listenformer outperforms other existing methods on most metrics across three attitudes, with particularly outstanding performance on the $\mathcal{D}_{ood}$ set. Specifically, it shows improvements of 3.75, 2.20, and 0.39 on average results of angle, expression, and translation coefficients, respectively. This could be attributed to the enhanced robustness of our non-autoregressive training and inference method, as well as the effectiveness of the proposed cross-modal fusion method in capturing representative information within audio and motion features.

Meanwhile, various video-level metrics are displayed in Tab. 2. Listenformer achieves the best performance in SSIM, PSNR, and CSIM, with improvements of 0.03, 1.07, and 0.05, respectively. However, it slightly lags behind the state-of-the-art model MFR-Net in CPBD and FID. It could be due to the fact that MFR-Net has made improvements to the rendering model, giving it a certain advantage in facial reconstruction. Improving the rendering model of Listenformer is also one of our future research directions.

*4.2.2 L2L dataset.* Tab. 3 presents the feature-level results of our proposed method and other existing methods on the L2L dataset. One can see that the proposed method outperforms all state-of-the-art approaches, which supports that the proposed non-autoregressive

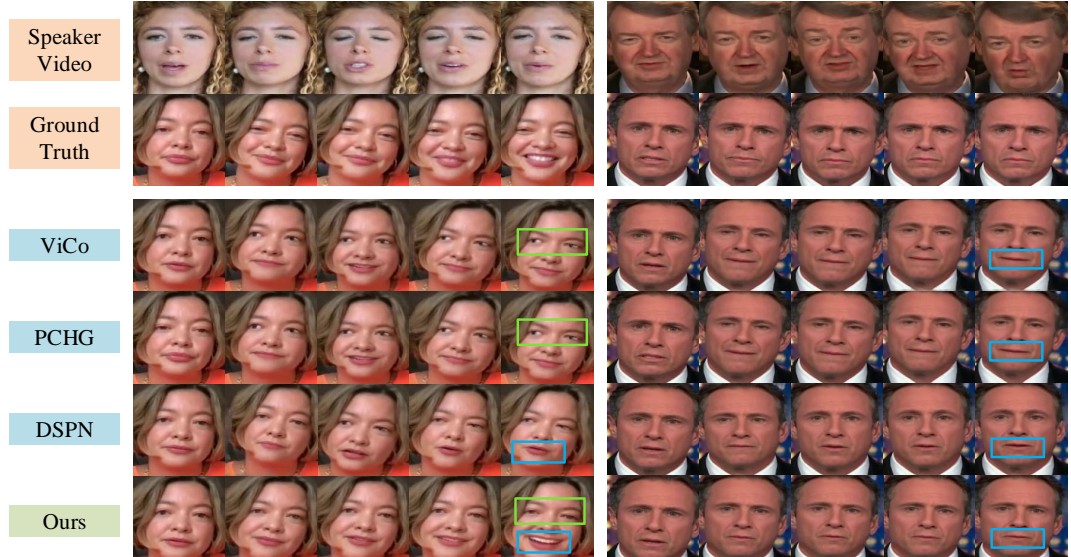

**Figure 5: Snapshots of the generated listener head videos (left: positive listener, right: neutral listener).**

**Table 4: User study results on ViCo dataset.**

| Method | ON ↑ | MD ↑ | IP ↑ | Sync ↑ | AT ↑ |
|---|---|---|---|---|---|
| ViCo [57] | 12.2% | 20.2% | 14.7% | 15.6% | 40.7% |
| PCHG [21] | 13.3% | 16.9% | 12.9% | 14.2% | 38.9% |
| DSPN [54] | 13.1% | 15.8% | 17.1% | 12.9% | 38.7% |
| Ours | **58.1%** | **46.4%** | **55.3%** | **58.0%** | **43.2%** |

**Table 5: Ablation study for fusion methods in ListenFormer tested on ViCo and L2L datasets.**

| Fusion Method | ViCo | | | L2L | |
|---|---|---|---|---|---|
| | PSNR ↑ | FID ↓ | CSIM ↑ | L1 ↓ | FD ↓ |
| Concat | 18.82 | 25.24 | 0.62 | 10.06 | 12.90 |
| Motion-dominated | 18.52 | 25.25 | 0.62 | 10.22 | 12.94 |
| Audio-dominated | **18.89** | **24.52** | **0.63** | **9.27** | **12.90** |

ListenFormer also exhibits significant advantages in modeling head motions and facial expressions on large datasets.

## 4.3 Qualitative Evaluation

To qualitatively evaluate different methods, we provide the responsive listening head frames generated by the proposed method and other methods in Fig. 5. We can see that Listenformer provides a reasonable response, which may not align entirely with ground truth but remains generally consistent. Both ViCo and PCHG struggle to maintain accurate identity information, specifically in (a), where ViCo and PCHG model eye movements unnaturally, and in (b), where generated listeners consistently maintain a weird smile. Although DSPN doesn't exhibit the aforementioned glaring shortcomings, it lacks sensitivity in capturing the positive attitude in (a) and neutral attitude in (b). Conversely, our approach ensures the preservation of accurate identity information without visible artifacts. Furthermore, the generated videos present more natural facial expressions and more precise attitude conveyance. Please watch the supplementary video for the dynamic comparison.

## 4.4 User Study

We invite 15 people to evaluate the generated listening head videos of our method with the other three methods. Each generated video along with its corresponding speaker's video is concatenated into the same video for presentation. 30 videos from the ViCo dataset

are involved in this user study with human measures in overall naturalness (ON), motion diversity (MD), identity preserving (IP), speaker-listener synchronization (Sync), and attitude matching (AT). Except for attitude matching, which offers a choice between positive, negative, and neutral, the remaining four options are selected as the best among the four methods. The results from all participants are averaged and listed in Tab. 4. ListenFormer achieves the best performance among all subjective measures, especially in terms of motion diversity, identity preservation, and synchronization. This verifies the capability of our method in generating diverse and natural listening head videos.

## 4.5 Ablation Study

*4.5.1 Effect of the audio-motion fusion module.* In this section, we conduct experiments to compare three different fusion methods. "Concat" refers to directly concatenating audio and motion features. "Audio-dominated" denotes the proposed AMAF method in Section 3.4, while "motion-dominated" involves swapping the positions of audio and motion features in the AMAF module. Tab. 5 presents the performance of three methods on two datasets. It can be observed that the "audio-dominated" method outperforms the other two methods in all metrics. This not only indicates the effectiveness of the proposed audio-motion attention fusion method but also suggests that the speaker's audio may be more crucial than motion for LHG, as it can convey more information, such as semantics.

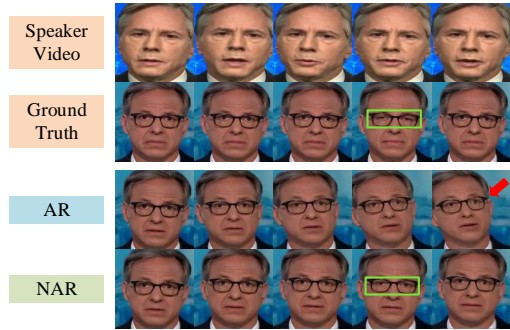

**Figure 6: Qualitative results of ListenFormer with different training methods.**

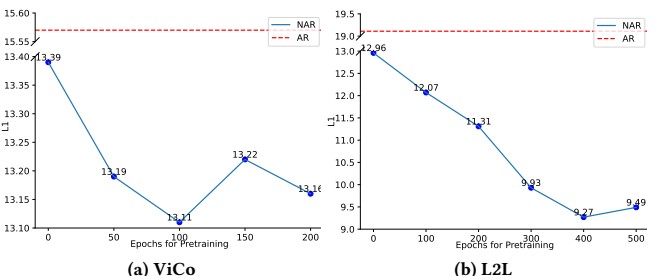

**Figure 7: L1 distance of ListenFormer trained under different pre-training epochs on ViCo and L2L datasets.**

*4.5.2 Effect of the training method.* Fig. 6 respectively illustrates the generation results of ListenFormer trained using autoregressive (AR) and non-autoregressive (NAR) methods (two-stage training). It is evident that the generated video of the autoregressive model exhibits an abnormal slant in the head, which is difficult to correct in subsequent inference steps. The non-autoregressive model consistently maintains stable head movement throughout the whole generated video. This demonstrates that non-autoregressive methods can significantly alleviate the inherent issue of error accumulation in autoregressive methods. Moreover, blinking and other motions also appear in generated videos of the NAR model, indicating that the NAR model retains excellent motion diversity due to the two-stage training method.

Fig. 7 specifically demonstrates the L1 results of models trained with different pre-training epochs for the expression and pose coefficients. On the ViCo dataset, the model achieves optimal performance with 100 pre-training epochs, while on the L2L dataset, the model performs best with 400 pre-training epochs. This may be attributed to the larger volume of data in the L2L compared to the ViCo dataset. Overall, the non-autoregressive method significantly outperforms the autoregressive method, even without the pre-training stage.

*4.5.3 Effect of the decoding method.* We conduct experimental comparisons of three different decoding methods mentioned in Section 3.5. Fig. 8 displays frames selected from a 26-second video clip spanning 5 to 15 seconds. It is evident that the all-in method, when inferred beyond the length of the training clips (3 seconds), leads

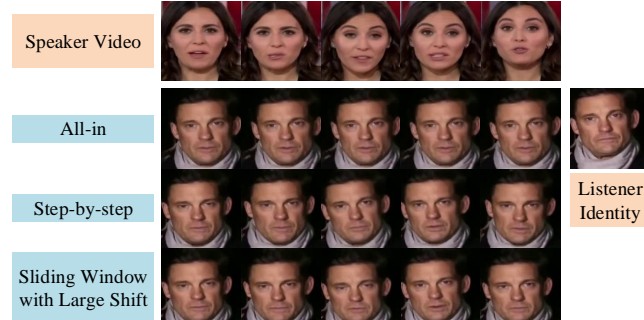

**Figure 8: Qualitative results of ListenFormer with different decoding methods.**

to static facial expressions and induces slight but rapid back-and-forth head movements. This is due to that the sinusoidal positional encoding fails to capture the modeling of position information for extended lengths. For the step-by-step method, although the generated facial expressions are no longer static, there are more pronounced back-and-forth head movements. As mentioned in Section 3.5, this may be attributed to the fact that for non-autoregressive ListenFormer, step-by-step inputs do not necessarily yield coherent results. The step-by-step approach introduces significant temporal jitters in the predictions, resulting in a visibly less smooth appearance. In comparison, our proposed method offers several advantages. On the one hand, the utilization of a sliding window helps to overcome the limitations associated with sinusoidal positional encoding for length extrapolation in decoding phase. On the other hand, the utilization of a large shift ensures that the generated frames do not exhibit jitters within the window. As a result, our method achieves superior visual quality compared to the other two methods. Additionally, it also leads to significant savings in computational resources according to Section 3.5.

## 5 CONCLUSION

We introduce a novel transformer-based model for the responsive listening head generation task. Our proposed Listenformer achieves non-autoregressive inference through teacher-forcing pre-training and input-changed fine-tuning stage, ensuring consistency between training and inference prediction modes. Additionally, to provide more accurate responses to the speaker inputs, an audio-motion attention fusion method is proposed, which better captures the audio-motion correlation information in the speaker's signals. To further enhance performance, we propose a sliding window with a large shift approach to address infinite-length inference scenarios, which performs well in terms of both effectiveness and computational efficiency. Qualitative and quantitative experiments have validated the superiority of our method over other state-of-the-art methods in generating high-quality listening head responses.

**Limitations:** The renderer and transformer are treated as independent components in our proposed method. In the future, we plan to explore joint optimization of these two components. Furthermore, we consider abandoning the rendering model and applying our method to 2D-based generation approaches.

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
