# OpenReview forum: "ListenFormer: Responsive Listening Head Generation with Non-autoregressive Transformers"
_acmmm.org/ACMMM/2024/Conference — MM2024 Poster_

### Official Review · Reviewer_VJ6p · 2024-05-22

**Rating:** 4
**Confidence:** 1

**Summary:**

This paper proposes an audio-motion attention fusion module, which improves the correlation of audio and motion features for accurate response, for listening head generation.

**Strengths:**

1. The paper clearly describes each and every module of the proposed system.

2. The experiments have thorough ablation highlighting the usefulness of each component.

3. The ideas described in the paper appear novel.

**Limitations:**

1. Lack of reproducibility. No source code was provided.

2. It would be useful to explain the evaluation metrics? Specifically what each of them capture.

**Suitability:**

3

---

### Official Review · Reviewer_6RqT · 2024-05-23

**Rating:** 4
**Confidence:** 2

**Summary:**

The paper generates a listening head video based on speaker's audio and video as well as reference listener's image. It addresses limitations in existing autoregressive methods used in human-robot interactions by proposing a two-stage training approach that prevents error accumulation and improves generation speed. It integrates an audio-motion attention fusion module to better correlate audio and motion features, enhancing the model’s ability to generate realistic listener head videos.

**Strengths:**

- The model is evaluated extensively on the ViCo and L2L datasets, demonstrating improvements over state-of-the-art methods in several metrics. The evaluation includes a user study to validate model effectiveness in realistic scenarios.

- The paper introduces a non-autoregressive framework of ListenFormer over traditional autoregressive models

**Limitations:**

-  The two-stage training process, while effective, introduces additional complexity in training the model, requiring fine tuning.
- The paper does not state if the model can run in real time.
- The supp. video does not show generated videos with negative attitude.

**Suitability:**

3

---

### Official Review · Reviewer_Enwp · 2024-05-25

**Rating:** 4
**Confidence:** 2

**Summary:**

The paper presents ListenFormer, a novel transformer-based model designed for generating responsive listening head videos. The primary contribution is the development of a non-autoregressive approach combined with an innovative audio-motion attention fusion module. This model effectively integrates cross-modal features and employs a sliding-window decoding method to address extrapolation limitations of transformers. Experiments on the ViCo and L2L datasets demonstrate the model's good performance in visual naturalness, generation diversity, and speaker-listener synchronization.

**Strengths:**

- Innovative Approach: The paper introduces a non-autoregressive transformer-based model and a novel audio-motion attention fusion module, showcasing significant advancements in responsive listening head video generation.
- Extensive Experimental Validation: The experiments are thorough, including quantitative and qualitative evaluations, user studies, and ablation studies, all of which validate the effectiveness of the proposed method.
- Clarity and Presentation: The paper is well-written and structured, making complex concepts accessible. Figures and tables are used effectively to enhance understanding.

**Limitations:**

- Reproducibility

**Suitability:**

3

---

### Official Review · Reviewer_8JLj · 2024-06-10

**Rating:** 3
**Confidence:** 2

**Summary:**

The authors propose Listenformer to predict the future listener heads based on the speaker inputs, a reference listener image and the past outputs. The experiments are conducted on the ViCo and L2L datasets. The main concern is the problem motivation of listening head generation. Compared with talking head generation, it is better for the authors to explain the motation of the problem of listening head generation. The second concern is the technical contribution of this paper, the audio-motion attention fusion method is not very novel and proposed non-auoregressive decoder is also not very novel.

**Strengths:**

The authors try the new problem of listener head generation.

**Limitations:**

1. The motivation of listener head generation is not very clear.
2. The technical contributions of proposed non-autoregressive decoder with audio-motion attention fusion is not very novel.
3. The dataset size of ViCo and L2L is not large enough, compared with talking head generation problem.

**Suitability:**

2

---

### Meta-Review · Area_Chair_J6iX · 2024-07-01

**Recommendation:** Accept (Poster)
**Confidence:** 4

**Metareview:**

this paper investigates the responsive listening head generation and introduces a new approach, ListenFormer, to take full advantage of the temporal modeling capability of transformer. experiments on multiple datasets showcase competitive performance and ablation study verifies the effectiveness of the proposed method.

the paper initially receives ratings of BR, BA, BA, BA. while there were concerns on clarity and novelty, other reviewers also praised this paper for its novel approach, the clear introduction of the proposed method, and thorough experiments. in rebuttal, the authors provided further clarification on some of the issues, and the overall ratings improved to BR, BA, WA, BA.

overall, the AC recommends Accept. since the author clearly introduced their new approach and findings, and verified them via thorough experiments, the AC believes it is worth sharing with the community. nonetheless, the authors are strongly encouraged to further improve their final manuscript based on the feedback from the reviewers.